# Gender Differences and Relationship of 2D:4D-Ratio, Mental Toughness and Dark Triad Traits among Active Young Adults

**DOI:** 10.3390/biology11060864

**Published:** 2022-06-05

**Authors:** Seyed Hojjat Zamani Sani, Dena Sadeghi-Bahmani, Zahra Fathirezaie, Mohammad Taghi Aghdasi, Kosar Abbaspour, Georgian Badicu, Serge Brand

**Affiliations:** 1Physical Education and Sport Sciences Faculty, University of Tabriz, Tabriz 5166616471, Iran; zahra.fathirezaie@gmail.com (Z.F.); mt.aghdasi@yahoo.com (M.T.A.); kosar.abbaspour@gmail.com (K.A.); 2Department of Psychology, Stanford University, Stanford, CA 94305, USA; bahmanid@stanford.edu; 3Sleep Disorders Research Center, Kermanshah University of Medical Sciences, Kermanshah 6719851115, Iran; Serge.Brand@upk.ch; 4Department of Physical Education and Special Motricity, Faculty of Physical Education and Mountain Sports, Transilvania University of Braşov, 500068 Braşov, Romania; georgian.badicu@unitbv.ro; 5Adult Psychiatric Clinics (UPKE), Center for Affective, Stress and Sleep Disorders (ZASS), University of Basel, 4002 Basel, Switzerland; 6Department of Sport, Exercise and Health, Division of Sport Science, University of Basel, 4052 Basel, Switzerland; 7Substance Abuse Prevention Research Center, Kermanshah University of Medical Sciences, Kermanshah 6714869914, Iran; 8School of Medicine, Tehran University of Medical Sciences (TUMS), Tehran 1417466191, Iran

**Keywords:** 2D:4D-ratio, dark triad, mental toughness, gender, active young adults

## Abstract

**Simple Summary:**

There is evidence that the exposure to more testosterone and less estrogen hormones before birth, that is, in utero, has a physiological impact on the relation between the index finger (2D) and the ring finger (4D). This relation is called the 2D:4D-ratio. A lower 2D:4D-ratio mirrors a longer ring finger (4D), relative to the index finger (2D); a higher 2D:4D-ratio mirrors a shorter ring finger (4D), relative to the index finger (2D). A higher exposure to testosterone and a lower exposure to estrogen hormones are associated with a lower 2D:4D-ratio. Further, a higher exposure to androgen hormones is associated with higher masculine traits. In the present study, we assessed 460 physically active young male and female adults. We showed that (1) males had lower 2D:4D ratios than females. Next, (2) active females and males had similar personality and mental toughness traits. (3) Females with more “dark” personality traits had a higher ability to perform consistently under stress and pressure; such females were mentally tougher. (4) Males with lower 2D:4D-ratios were mentally tougher. Overall, there is evidence that prenatal exposure to androgen hormones and personality traits in adulthood appear to be related, though, in a different fashion between male and female adults.

**Abstract:**

There is consistent evidence that prenatal exposures to higher testosterone and lower estrogen concentrations during the first trimester of embryonal and fetal development are associated with a lower 2D:4D-ratio, which is to say: The index finger (2D) is shorter, compared to the ring finger (4D). Compared to non-active, athletes show lower 2D:4D ratios. However, athletes also report specific personality traits such as mental toughness, assertiveness, and competitive behavior. Here, we tested if 2D:4D-ratios were related to specific personality traits. We further investigated possible gender differences. A total of 460 active young adults (mean age: 24.81 years; 67% females) completed a series of self-rating questionnaires covering sociodemographic information and traits of the dark triad and mental toughness. Participants also provided a scan of their right palm hand to measure and calculate 2D:4D-ratios. *t*-tests, Pearson’s correlations, and multiple regression analysis were performed to analyze data. Compared to male participants, female participants had a higher 2D:4D-ratio. Female and male participants did not differ as regards dark triad traits and mental toughness traits. Irrespective of gender, and based on correlational computations, 2D:4D-ratios were unrelated to the dark triad (DT) and mental toughness (MT) scores. Higher DT scores were modestly associated with higher MT scores among females, but not among males. Lower 2D:4D-ratios were associated with higher constancy scores and the male gender. The constancy and male gender appeared to be associated with lower 2D:4D-ratios.

## 1. Introduction

Among highly skilled elite performers, it appears that striving for excellence is associated with a combination of physical and psychological factors [1,2]. Specifically, there is some evidence that individuals exposed prenatally to higher testosterone and lower estrogen concentrations might have a better physical performance as children [2,3,4], adolescents [5,6] (but see also [7] for opposite results), and adults [8,9,10,11,12,13,14,15]. Given this background, the following theoretical foundations and hypotheses were examined.

The relation between the lengths of the index finger (2D) and the lengths of the ring finger (4D) is called the 2D:4D-ratio [16]. A lower 2D:4D-ratio equals a shorter index finger, relative to the ring finger; a higher 2D:4D-ratio equals a longer index finger, relative to the ring finger. There is consistent evidence from animal [17,18,19] and human studies [16] that prenatal exposure to higher testosterone concentrations and lower estrogen concentrations during the first trimester of embryonal and fetal development leads to a lower 2D:4D-ratio. Evidence comes from studies on congenital adrenal hyperplasia (CAH), a neuroendocrine dysregulation associated with an increase in the size of the fetal adrenal glands and an elevated level of fetal androgens. A meta-analysis on this topic showed that compared to typically developing children, children with CAH had lower 2D:4D-ratios, and thus more “masculinized” 2D:4D-ratios [20]. Further, when investigating 2D:4D-ratios among professional elite performers, it appears that a lower 2D:4D-ratio was associated with higher objectively assessed performance [16]. There is further agreed that a lower 2D:4D-ratio, understood as a proxy of prenatal higher testosterone and lower estrogen concentrations, does not mechanically and linearly lead to higher physical performance or even to “better” personality traits and, thus, to a more goal-oriented assertive behavior. Rather, the prenatally higher testosterone concentration appears to enable the organism to outperform in case of necessity [16], that is, under challenging conditions. Further, such patterns were consistently observed among males, and inconsistently among females, always when compared to non-professionally exercising counterparts [16]. Last, there is sparse evidence that a lower 2D:4D-ratio was associated with higher Dark Triad traits [21]. Given this background, two further aims of the present study were to compare the 2D:4D-ratio between male and female young adults, and to investigate if their 2D:4D-ratios were associated with Dark Triad traits.

Next, there is sufficient evidence that, compared to females, males have lower 2D:4D-ratios [16,22,23]. In this view, a lower 2D:4D-ratio was associated with higher athletic performance, though, more so in males, compared to females [20].

Next, and based on previous findings, a lower 2D:4D-ratio was associated with more assertive behavior. Typically, assertive behavior was assessed with dimensions of typical male occupations such as car mechanic, builder, carpenter, electric engineer, or inventor [24]. However, against expectations, a lower 2D:4D-ratio was not observed among male firefighters, when compared to a normal male population [25]. Further, in both men and women, a lower 2D:4D ratio was statistically significantly associated with higher scores for aggression, thrill and adventure seeking, and sensation-seeking [26]. Studies also showed that assertive behavior and high performance in sports and exercising were associated with higher MT traits [27,28,29], while for DT, to our knowledge, only two studies have addressed this research question so far [30]: Among 341 adults, higher DT traits were associated with higher vigorous physical activity levels. Among 189 athletes, higher DT traits predicted higher scores in basketball tasks [31].

Focusing on psychological traits to explain high performance among elite athletes, both the Dark Triad (DT) and Mental Toughness (MT) demand particular attention.

The Dark Triad (DT) comprises the personality traits of psychopathy, narcissism, and Machiavellianism. As such, “dark” refers to a socially highly discouraged personality framework and behavior [32,33]. Nevertheless, it appeared that DT traits were directly and indirectly associated with higher sports performance [31,34,35]. Further, DT traits were associated with more masculinity traits and fewer femininity traits among two samples of college students [36].

Besides DT traits, the concept of mental toughness [37,38,39,40] has attracted scientific attention. Mental toughness refers to cognitive–emotional personality traits consisting of the Four Cs: namely Commitment to one own’s defined goals, Confidence in oneself and others; Challenge in that changes are perceived as possibilities to improve, and Control over one own’s life and behavior [37]. Compared to the concept of resilience [41,42,43], it appears the MT offers a broader range of well-established and validated self-rating questionnaires [29,44,45,46]. Further, there is extant research among healthy adolescents [47,48,49,50], healthy adults [27,38,51,52,53,54,55,56,57,58,59,60,61,62], and adult individuals with multiple sclerosis [63,64] to show that higher MT scores were related to higher physical activity levels.

To our knowledge, only two studies investigated the associations between DT and MT [30]. A total of 341 young adults (mean age; 29 years; 51% females) completed self-rating questionnaires on DT and MT. Participants scoring high in MT traits reported also higher scores on all traits of DT of Machiavellianism, narcissism, and psychopathy. Gender differences were not observed. Higher scores on narcissism were associated with higher MT scores among a larger sample of 762 adults [34]. Given this background, the next aims of the present study were to investigate if the previous pattern of association could be replicated, and if and to what extent 2D:4D-ratios were associated with MT and DT.

To summarize, there is evidence of the prenatal formation of the 2D:4D-ratio; further, a lower 2D:4D-ratio was associated with more masculine traits, male gender, and higher physical activity indices. Next, higher physical activity indices were associated with personality traits of mental toughness (MT) and Dark Triad (DT). 

The present study aimed to investigate the relations between 2D:4D-ratios and the psychological dimensions of DT and MT among active female and male adults. We further investigated gender differences. 

Based on the scientific background described above, the following four hypotheses were formulated.

First, following previous findings [16,24], we expected a lower 2D:4D-ratio in male participants, compared to female participants (HYP1). Second, we predicted that a lower 2D:4D-ratio was associated with higher DT traits [21] (HPY2).

Third following two previous studies [30,34], we expected that MT scores and DT traits were associated (HYP3).

Fourth, following Jonason and Davis [36], we predicted that male participants reported higher DT scores, compared to their female counterparts (HYP4).

We claim that the present study had the potential to shed some more light on the sophisticated and intertwined associations between 2D:4D-ratios, MT, and DT, always separately for active male and female young adults.

## 2. Methods

### 2.1. Procedure

Students participating in university sports competitions from the Tehran, Shiraz, and Tabriz Universities (Tehran, Shiraz, and Tabriz, Iran) were approached to participate in the present study. They were fully informed about the aims of the study and the secure and anonymous data handling. Thereafter, participants signed the written informed consent. Next, participants completed a booklet of self-rating covering sociodemographic information, MT, and DT. Last, participants provided a scan of the palm of the right hand to measure the lengths of the digit and ring finger and calculate the 2D:4D-ratio (see details below). The local ethics committee of the Sports Sciences Research Institute of Iran (SSRII; Tehran, Iran) approved the study (IR.SSRI.REC.1400.1321), which was performed in accordance with the current and seventh edition [65] of the Declaration of Helsinki. 

### 2.2. Measures

#### 2.2.1. Sociodemographic Characteristics

Participants included athletic students from university teams in various sports who routinely practiced their sports. They reported on their gender at birth (male; female) and age (in years) and their participation in team or individual sports.

#### 2.2.2. Dark Triad

Participants completed the Farsi version [66] of the Dirty Dozen concise measurement of the Dark Triad [67]. The questionnaire consists of 12-items. Typical items are “I have used deceit or lied to get my way”; “I tend to manipulate others to get my way”; or “I tend to be unconcerned with the morality of my actions”. Answers are given on 5-point Likert-type scales ranging from 1 (=strongly disagree) to 5 (=strongly agree), with higher sum scores reflecting a more pronounced tendency to DT traits (Cronbach’s alpha = 0.72). 

#### 2.2.3. Mental Toughness

Participants completed the Farsi version [68] of the Sport Mental Toughness Questionnaire (SMTQ) [69]. The questionnaire consists of 14 items and loads on the subscales Confidence, Constancy, and Control. Typical items are: “I interpret threats as positive opportunities” (Confidence); “I give up in difficult situations” (Constancy); “I’m overcome with self-doubt” (Control). Answers are given on 4-point Likert scales, ranging from 1 (=not at all true) to 4 (=very true), with some items being reversed coded. After recording, a higher sum score reflected a higher mental toughness (Cronbach’s alpha = 0.67). 

#### 2.2.4. Digit Ratio Index 

The 2D:4D ratio is the most studied digit ratio and is calculated by dividing the length of the index finger of a given hand by the length of the ring finger of the same hand. Vernier Caliper with a resolution of 0.01 mm was used for the indirect measurements of the 2nd (index) and 4th (ring) fingers from the scan of the right palm hand of participants. Mean digit ratio for women was 0.99, SD = 0.03; men was 0.98, SD = 0.04. 

#### 2.2.5. Statistical Analysis

To test the first (compared to females, males have lower 2D:4D-ratios; HYP1) and fourth hypothesis (males reported higher DT scores than females; HYP4), a series of *t*-tests was performed. To answer the second (lower 2D:4D-ratios were associated with higher DT traits; HYP2) and third hypothesis (higher MT scores and higher DT traits were associated; HYP3), a series of Pearson’s correlations were performed. 

For *t*-tests, besides *p*-values, we reported effect sizes (Cohen’s ds) with the following cut-off values [70,71,72]: d < 0.19 = trivial effect size; 0.20 < d < 0.49 = small effect size; 0.50 < d < 0.79 = medium effect size; d > 0.80 = large effect size.

For Pearson’s correlation coefficients, the following cut-off values were reported: r < 0.09 = trivial correlation; 0.10 < r < 0.29 = small correlation; 0.30 < r < 0.49 = medium correlation; r > 0.50 = large correlation.

For the multiple regression analysis (following others [73,74], preliminary conditions to perform a multiple regression analysis were generally met: the sample size (*n*; females = 308; *n*; males = 152) was >100; the number of predictors × 10 should not be greater than the sample size (here: 3 × 10 = 30 < 308; 152); predictors should sufficiently explain the dependent variable (R and R^2^); and the Durbin–Watson coefficient should be between 1.5 and 2.5, indicating that the residuals of the predictors were independent of each other. Last, the variance inflation factors (VIF) to test multicollinearity should be 1 < VIF < 10.

## 3. Results

### 3.1. General Information

A total of 460 individuals (308 females; 67%; mean age: 24.56 years (SD = 1.96); 152 males; 33%; mean age: 25.34 years (SD = 2.18), participated in the study.

### 3.2. 2D:4D-Ratios, Dark Triad, and Mental Toughness between Male and Female Participants

Table 1 provides the descriptive and statistical overview of dimensions of 2D:4D-ratios DT and MT traits between male and female participants.

Compared to male participants, female participants had a higher 2D:4D-ratio (*p* = 0.001; d = 0.32; *small effect size*). 

For DT, compared to male participants, female participants reported lower overall scores for DT (*p* = 0.029, d= 0.21; *small effect size*), and psychopathy (*p* = 0.008, d = 0.26; *small effect size*), but not on Machiavellianism (*p* = 0.064, d= 0.18; *trivial effect size*), and narcissism (*p* = 0.98, d = 0.002; *trivial effect size*).

For MT, compared to male participants, female participants did not report a different overall score, or different scores on confidence and control (ps > 0.1; 0.03 ≤ ds ≥ 0.11; *always trivial effect sizes*), though, descriptively lower scores for constancy (*p* = 0.024; d = 0.22; *small effect size*). 

### 3.3. Correlations between 2D:4D-Ratio, and Traits of Mental Toughness and Dark Triad among Female Participants (n = 308)

Table 2 provides the correlational associations (Pearson’s coefficients) between the 2D:4D-ratio, and traits of DT and MT among female participants.

2D:4D-ratios and DT traits were unrelated; thus, the second hypothesis (HYP2) was rejected. A lower 2D:4D-ratio was associated with higher constancy (small effect size). 

Higher DT traits (sum score) were associated with higher scores of Machiavellianism, psychopathy, and narcissism (large effect sizes), and with lower constancy and control (small effect sizes). For all other dimensions, correlation coefficients were trivial.

Higher scores of Machiavellianism were associated with higher psychopathy and narcissism (small to medium effect sizes), and lower confidence (small effect size). For all other dimensions, correlation coefficients were trivial.

Higher scores of psychopathy were associated with lower dimensions of mental toughness, confidence, constancy, and control (small effect sizes).

Higher scores of narcissism were associated with higher mental toughness, confidence, and control (small effect sizes).

Mental toughness traits were highly interrelated (medium to large effect sizes. Exception: Constancy and control were not associated.

Overall, 2D:4D-ratio was not associated with MT and DT traits; some higher DT traits were both negatively and positively associated with MT traits, though, effect sizes were small.

### 3.4. Correlations between 2D:4D-Ratio, and Traits of Mental Toughness and Dark Triad among Male Participants (n = 152)

Table 3 reports the correlational associations (Pearson’s coefficients) between the 2D:4D-ratio, and traits of mental toughness and dark triad among male participants.

A lower 2D:4D-ratio was associated with higher constancy (small effect size). 

2D:4D-ratios and DT traits were unrelated. 

Higher DT traits (sum score) were associated with higher scores of Machiavellianism, psychopathy, and narcissism (large effect sizes), while no associations were observed with MT traits. 

Higher scores of Machiavellianism were associated with higher psychopathy and narcissism (medium effect sizes), while no associations were observed with MT traits.

Higher scores of psychopathy were associated with higher narcissism (medium effect size), while no associations were observed with MT traits.

Higher scores of narcissism were associated with higher mental toughness, confidence, and control (small effect sizes).

Mental toughness traits were highly interrelated (medium to large effect sizes). Exception: Constancy and control were modestly associated.

Overall, 2D:4D-ratio was not associated with MT and DT traits; some higher DT traits were not associated with MT traits.

To summarize the associations observed among female and male participants, as a general pattern, 2D:4D-ratios were unrelated to DT and MT scores, and DT scores were modestly associated with MT scores among females, but not among males. 

### 3.5. Gender, Mental Toughness Traits, and Dark Triad Traits to Predict 2D:4D-Ratios

A multiple regression analysis was performed to predict 2D:4D-ratios; predictors were gender, traits of mental toughness, and dark triad. Table 4 reports the results. 

Higher scores of constancy predicted lower 2D:4D-ratios; simply put: Participants with higher scores of constancy had lower 2D:4D-ratios. Next, the male gender was associated with lower 2D:4D-ratios. Dimensions of MT and DT were excluded from the equation, as they did not reach statistical significance. 

We answered the third research question (RQ3): Both a higher score of constancy and male gender predicted lower 2D:4D-ratios, though, R (0.247) and R^2^ (0.061) were modest: Both constancy scores and male gender predicted 6.1% of the variance of the 2D:4D-ratios; accordingly, 93.9% of the variance of the 2D:4D-ratios remained unexplained.

## 4. Discussion

The present study aimed to investigate the associations between 2D:4D-ratios, dark triad, and mental toughness traits among physically active female and male young adults. The key findings were as follows: First, compared to male participants, female participants showed higher 2D:4D-ratios. Second, both female and male participants did not differ as regards mental toughness traits, though, compared to males, females reported lower scores for DT traits. Third, 2D:4D-ratios were unrelated to DT and MT scores, and fourth, DT scores were modestly associated with MT scores among females, but not among males. Fifth, lower 2D:4D-ratios were associated with higher constancy and male gender. 

With the first hypothesis (HYP1) we assumed that relative to their female counterparts, male participants showed lower 2D:4D-ratios, and data did confirm this. Thus, we replicated what has been observed before [16,24]. The theoretical basis is that compared to females, males are exposed to higher testosterone and lower estrogen concentrations during the first trimester of embryonal and fetal development. The novelty of the result is that such a pattern has been observed among active young adults in Iran.

With the second hypothesis (HYP2), we predicted that a lower 2D:4D-ratio was associated with higher DT traits, though data did not confirm this. Given this, we did not confirm what has been observed before [21]. A closer comparison of the present data compared with those of Borráz-León et al. [21] revealed that they observed a correlation coefficient of r = 0.21 between the 2D:4D-ratio of the left hand and narcissism among 119 healthy college students, while in the present study, 2D:4D-ratios of the right hand of active young adults were assessed; as such, systematic biases of samples and assessment measured might explain the mismatch of results. However, the relation between 2D:4D and spatial abilities (i.e., cognitive functions) has been confirmed in the previous research [75]. 

With the third hypothesis (HYP3) we expected that MT scores and DT traits were associated; this was confirmed for female, but not for male participants. As such, we could partly confirm what has been observed in two previous studies [30,34]. 

With the fourth hypothesis (HYP4) we predicted that male participants reported higher DT scores, compared to their female counterparts, and this was confirmed. Thus, we could confirm previous data [36] and expand upon them, in that such patterns were observed also among a larger group of Iranian active young adults. 

We further asked if females reported lower MT scores, compared to male participants, and the answer was no. Previous studies showed that MT scores were associated with higher physical activity and exercise scores [30,47,48,51,63,64,76,77,78] and that such physical activity and exercise levels were associated with dimensions of psychological functioning. Given this, it is conceivable that the sample of active young adults was too homogeneous in their physical activity levels and psychological functioning, such to preclude higher statistical variance and thus a more meaningful pattern. In this view, the standard deviations for the DT and MT scores reported in Table 1 might suggest that this could be the case. Given such low variances, the odds to observe “interesting” correlation coefficients also decreased. A further possibility is that there was no association between the two personality dimensions.

In addition, we asked if lower 2D:4D-ratios were associated with higher MT scores, and the answer was yes, though among female, but not among male participants, and just for the dimension Constancy.

Constancy is understood as the attitude to remain persistent and to strive for one’s goals irrespective of and despite difficult situations and obstacles. As such, it appears that constancy might be best mirrored as hardiness [79,80] and resilience [41,81,82]. The present data expand upon the current literature in that we could show that lower 2D:4D-ratios, and thus higher testosterone and lower estrogen concentrations during the first trimester of embryonal and fetal development, were associated with the current personality trait of mental toughness. As such, we follow others [16] and claim that prenatal exposure to male sex steroids concentrations does not directly impact current behavior and personality traits, but enables the organism to (out-) perform in case of necessity. In such a context, constancy is understood as the cognitive–emotional appraisal to (out-)perform under challenging circumstances. 

Last, we asked, which factors (MT, DT, and gender) could more accurately predict 2D:4D-ratios; the answer was that the combination of male gender and higher constancy scores predicted lower 2D:4D-ratios, that is, higher exposure to testosterone during the prenatal and fetal development and the current attitude to (out-) perform in case of necessity. As such, the results of the regression analysis confirmed what has been observed and discussed above. 

Despite the novelty of the present study findings, several limitations should be considered. First, the cross-sectional nature of the study does not allow for drawing causal conclusions. A future and longitudinal study design might solve such an issue. Second, it is conceivable that further latent and unassessed psychological dimensions such as symptoms of depression and anxiety might have biased two or more dimensions in the same or opposite directions. Given this, future studies should assess participants’ psychological functioning more broadly and both via self- and experts’ ratings. Third, while the athlete status was an inclusion criterion, current physical activity and exercising patterns in terms of frequency, duration, and intensity were not assessed. Future studies should consider this confounder. Fourth, the introduction of a control group consisting of non-athletes would have allowed us to judge if, and to what extent, the present pattern of results is specific to adult athletes, or merely general to individuals in young adulthood.

## 5. Conclusions

Among active female and male young adults, 2D:4D-ratios as a proxy of prenatal exposure to higher testosterone and lower estrogen concentrations are associated with specific psychological dimensions such as DT and MT traits in a gender-specific fashion. Further, MT traits of constancy and male gender predicted lower 2D:4D-ratios. Lastly, while statistically significant gender differences were observed for 2D:4D ratio, DT, psychopathy, and constancy, effect sizes were small or trivial. Given this, the pattern of results should be interpreted with caution.

## Figures and Tables

**Table 1 biology-11-00864-t001:** Descriptive and inferential statistical overview of 2D:4D-ratios, dimensions of mental toughness (MT), and dark triad (DT) between female and male participants.

	Gender		Statistics	
	Female	Male	t (458)	Cohen’s d
N	308	151		
	M (SD)	M (SD)		
2D:4D-ratio	0.995 (0.037)	0.983 (0.036)	3.22 ***	0.32 (S)
Dark Triad				
Total score	28.95 (5.70)	30.23 (6.21)	2.19 *	0.22 (S)
Machiavellianism	7.01 (2.93)	7.56 (3.07)	1.98	0.18 (T)
Psychopathy	7.76 (2.44)	8.39 (2.25)	2.66 **	0.27 (S)
Narcissism	14.25 (2.80)	14.25 (3.03)	0.39	0.00 (T)
Mental toughness				
Total score	40.20 (5.54)	40.01 (5.54)	0.35	0.04 (T)
Confidence	17.61 (2.74)	17.31 (2.56)	1.14	0.11 (T)
Constancy	11.24 (2.36)	11.77 (2.29)	2.27 *	0.23 (S)
Control	11.18 (2.42)	10.94 (2.15)	1.07	0.11 (S)

Notes: * = *p* < 0.05; ** = *p* < 0.01; *** = *p* < 0.001. T = trivial effect size; S = small effect size.

**Table 2 biology-11-00864-t002:** Correlational associations (Pearson’s coefficients) between 2D:4D-ratio, mental toughness, and dark triad traits among female participants (*n* = 308).

			Dimensions						
Dimensions	2D:4D-Ratio	Dark Triad	Machiavellianism	Psychopathy	Narcissism	Mental Toughness	Confidence	Constancy	Control
2D:4D-ratio	-	0.02	−0.04	0.03	0.04	−0.10	−0.03	−0.20 **	0.02
Dark Triad		-	0.78 ***	0.64 ***	0.62 ***	−0.02	−0.05	−0.15 **	0.12 *
Machiavellianism			-	0.34 ***	0.22 ***	−0.05	−0.15 **	−0.02	0.07
Psychopathy				-	0.08	−0.23 *	−0.19 **	−0.25 **	−0.15 *
Narcissism					-	0.21 ***	0.20 **	−0.07	0.31 ***
Mental toughness						-	0.83 ***	0.68 ***	0.61 ***
Confidence							-	0.47 ***	0.37 ***
Constancy								-	0.10
Control									-

Notes: * = *p* < 0.05; ** = *p* < 0.01; *** = *p* < 0.001.

**Table 3 biology-11-00864-t003:** Correlational associations (Pearson’s coefficients) between 2D:4D-ratio, mental toughness, and dark triad traits among male participants (*n* = 152).

			Dimensions						
Dimensions	2D:4D-Ratio	Dark Triad	Machiavellianism	Psychopathy	Narcissism	Mental Toughness	Confidence	Constancy	Control
2D:4D-ratio	-	−0.02	−0.10	−0.02	0.11	−0.15	−0.09	−0.21 *	−0.03
Dark Triad		-	0.76 ***	0.71 ***	0.77 ***	0.12	0.06	0.04	0.11
Machiavellianism			-	0.36 ***	0.32 ***	0.04	0.01	0.04	0.02
Psychopathy				-	0.38 ***	0.00	0.06	−0.07	0.01
Narcissism					-	0.24 **	0.19 *	0.08	0.23 **
Mental toughness						-	0.80 ***	0.65 ***	0.75 ***
Confidence							-	0.27 **	0.53 ***
Constancy								-	0.21*
Control									-

Notes: * = *p* < 0.05; ** = *p* < 0.01; *** = *p* < 0.001.

**Table 4 biology-11-00864-t004:** Dimensions to predict 2D:4D-ratios; multiple regression analysis.

Dimension	Variables	Coefficient	Standard Error	Coefficient β	t	*p*	R	R2	Durbin-Watson	VIF
2D:4D-ratio	Constant	1.04	0.009	-	110.90	0.000	0.247	0.061	2.02	
	Constancy	−0.003	0.001	−0.198	4.339	0.001				1.011
	Gender	−0.010	0.004	−0.128	2.809	0.005				1.011

Notes: VIF = Variance Inflation Factor. Coding of gender: 1 = female; 2 = male.

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
