# Peer review of "Gender Differences and Relationship of 2D:4D-Ratio, Mental Toughness and Dark Triad Traits among Active Young Adults"

_biology, 2022, doi:10.3390/biology11060864_

Round 1

Reviewer 1 Report

Thank you for taking the time to address my initial comments. I have suggested some further (smaller revisions) that I believe will improve the quality of the paper.

Introduction:

Broadly speaking, the introduction still needs some work, as it is hard to read in places. I would suggest making it much more concise:

For example, your first sentence says:

“Among highly skilled elite performers, it appears that striving for 53 excellence is associated either with psychological factors such as 54 resilience [1], with higher physical factors, or, most likely, with the 55 combination of physical and psychological factors [2].”

Why not just say:

“Among highly skilled elite performers, it appears that striving for excellence is associated with a combination of physical and psychological factors [1, 2].”

Line 56: the back half of this sentence does not make sense. Please revise the part that says “have advantages in sport outperforming when they are children”

Line 94: change “Next, there is sufficient evidence that the 2D:4D-ratio is sexually dimorphic: Compared to females, males have lower 2D:4D-ratios” to “Next, there is sufficient evidence that , compared to females, males have lower 2D:4D-ratios”

Line 106: remove “Next, former”

Line 149 – 153: please remove this information, as it pertains to the methods.

Line 158: as per previous comments, I would suggest removing the research questions, as you already have provided aims and hypothesis.

Methods:

As above, I would suggest removing the research questions, which then do not need to be specifically discussed in the methods.

What do you mean by “data gathering and elaboration”? please revise.

Results:

Effect sizes should be reported as follows (P = .029; d = 0.21; small effect) please revise throughout.

Discussion:

As per above comments – too much time is spent talking about both the hypothesis and research questions. I would suggest removing the research questions to improve the readability of the paper.

Line 366: remove “descriptively; but not considering the effect sizes” – there was a significant difference, that is all you need here.

Line 376: This sentence does not make sense – please revise.

Line 387: please remove this sentence, as it is unnecessary.

Line 396: can you please expand on the following “as such, systematic biases of samples and assessment measured might explain the mismatch of results.” What do you mean by this statement?

Line 402: what do you mean by “when dealing with 2D:4D ratios”? – please revise.

Author Response

Dear Reviewer,

Thank you very much for all your kind efforts.

We have addressed all concerns raised by the Reviewers. Please see the detailed point-by-point-response attached as a separate file. 

Again, thank you very much for the care devoted to our manuscript. 

Reviewer 2 Report

This manuscript deals with an interesting topic.The backgroud is clearly described in the Introduction. I recommend to refer in addition to interesting related work in rats and healthy human subjects by Müller N. et al. (2018) 2D:4D  and spatial abilities: From rats to humans. Neurbiology of Learning and Memory 151: 85 -87. The definition of Dark Triad traits is given twice in lines 89 and 117 as well. Once is enough. Methods are timely and appropriate. Novel interesting findings are clearly presented. The Discussion is appropriate. Limitations are described.

Furthermore I suggest to cite the work by Müller et al, whereas this is just a recommendation.

Author Response

Dear Reviewer,

Thank you very much for all your kind efforts.

We have addressed all concerns raised by the Reviewers. Please see the detailed point-by-point-response attached as a separate file. 

Again, thank you very much for the care devoted to our manuscript. 

This manuscript is a resubmission of an earlier submission. The following is a list of the peer review reports and author responses from that submission.

Round 1

Reviewer 1 Report

Larger Comments:

The introduction is not particularly well written. Please make much more concise.

You suggest that this study is performed on athletes, but in the methods, you state that you simply recruited sports science students who play sport?  You also did not obtain a measure of their physical activity levels, and you have not provided any supporting evidence to suggest that they participate in sport at an acceptable level (i.e., are they really athletes?). Without this information, I think the term “athletes” is misleading, I would suggest changing to “active young adults” throughout the entirety of the manuscript, including the title.

There are many statistical analyses conducted in this paper. I am of the belief that you should conduct a post-hoc correction for multiple comparison to reduce the likelihood of type 1 errors, which may have occurred.

The discussion is very wordy, and overly complex. I don’t think you need to provide a discussion point for all your hypothesis/research aims. I would instead focus on the key findings and relate that back to the relevant literature.

Specific comments:

Abstract:

Line 29: can you provide the specific analysis conducted in the methods.

Line 33: remove the words “descriptively and”

Line 36: Remove “as a general pattern” – non-significant patterns don’t really matter here. Just report on the statistical results.

Line 37: need to state DT and MT in full first time (or identify that these relate to metal toughness/dark triad above)

Line 39: remove “that is, a higher exposure to testosterone during prenatal stage” as you have already described this

Line 41: I don’t think this is an appropriate conclusion – all you can say from your findings is that “Lower 2D:4D-ratios were associated with higher constancy scores of MT and male gender.”

Line 44: remove “and thus with a higher prenatal exposure to higher testosterone and lower estrogen concentrations during the first trimester of embryonal and fetal development.”

Introduction:

Line 51: remove “in general”

Line 54: Change “in specific” to “specifically”

Line 58: remove the aims from this section – you state them below

Line 63: no need to say (see below) throughout the introduction.

Line 71: this is poorly written – please revise.

Line 78: change “The” to “A” meta-analysis

Line 80: remove “and interstingly”

Line 103: remove this sentence, it should be in the aims or methods.

Section “the present study” – there is no need to provide both a research questions and a hypothesis. I would suggest removing the research questions and leaving the hypothesis.

Methods:

Can you please provide previously published reliability estimates for each of the measures used (moth questionnaire data, and 2D;4D ratio measures).

Line 229: Change “analytic plan” to “statistical analysis

Line 238: Change “we run a Pearson’s correlation” to “Pearson’s correlation was used”

Results:

Please remove any discussion of confirming / disproving hypothesis and research questions from the results section. Just report on the results, and highlight this in the discussion.

Can you please report the specific effect size value (Cohens d) for each of the findings you report in text).

Page 11: remove “which in turn are believed to be associated with a higher prenatal exposure to testosterone”

Discussion:

Where did the line numbers go?

Paragraph 1: please remove “as a proxy of prenatal higher testosterone and lower estrogen concentrations during the first trimester of embryonal and fetal development,”

Paragraph 1: this is way to wordy, please make the key findings more concise.

Paragraph 2: remove “Further, a closer look at Table 1 reveals that standard deviations of the variables were generally small. Though, large standard deviations are necessary to identify meaningful correlation coefficients.”

Limitations paragraph: please revise the sentence “so, strickly taken, multiple regression analyses might be questioned.” This is unclear – be more specific.

Conclusion:

Try and summarise your key findings here. At the moment is very vague and unclear.

Reviewer 2 Report

First of all, I would like to congratulate the authors for their efforts.

General comment.

The research is well designed but sloppy written. I would like to review it again after major revisions.

Special Comments

-It is not clear what is meant in the title.

-You need to revise the abstract. There are places with format inconsistency.

-The introduction must be rewritten.

Subheadings in this section break the flow. Please stream.

Line 66 Please remove.

Line 99 Please remove.

Line 107 Please remove.

Line 123 Please remove.

Line 127 Please remove.

Line 135 Please remove.

Line 167 Please remove.

-Method section is not repeatable for readers. I invite you to write more descriptive and detailed.

Line 206 Please provide more explanation under this heading.

-Line 229 Please provide more explanation under this heading.

-Line 233- 262 Rewrite please.

- I think the analytical plan sentence is not appropriate.

-Tables and results should be completely revised. There is no flow in the resultss section and sentences are broken. Tables are not formatted, rearrange them. Provide stability in the format.

-Please improve the conclusion section.